# Corneal Epithelial Regeneration: Old and New Perspectives

**DOI:** 10.3390/ijms232113114

**Published:** 2022-10-28

**Authors:** Alessia Nuzzi, Francesco Pozzo Giuffrida, Saverio Luccarelli, Paolo Nucci

**Affiliations:** 1Eye Clinic, IRCCS MultiMedica, 20099 Milan, Italy; 2Ophthalmology Residency Program, University of Milan, 20122 Milan, Italy; 3Ophthalmological Unit, Fondazione IRCCS Ca’ Granda Ospedale Maggiore Policlinico, 20122 Milan, Italy; 4Department of Biomedical, Surgical and Dental Sciences, University of Milan, 20122 Milan, Italy

**Keywords:** ocular stem-cell therapies, regenerative ophthalmology, gene therapy, extracellular vesicles, exosomes, miRNA

## Abstract

Corneal blindness is the fifth leading cause of blindness worldwide, and therapeutic options are still often limited to corneal transplantation. The corneal epithelium has a strong barrier function, and regeneration is highly dependent on limbal stem cell proliferation and basement membrane remodeling. As a result of the lack of corneal donor tissues, regenerative medicine for corneal diseases affecting the epithelium is an area with quite advanced basic and clinical research. Surgery still plays a prominent role in the treatment of epithelial diseases; indeed, innovative surgical techniques have been developed to transplant corneal and non-corneal stem cells onto diseased corneas for epithelial regeneration applications. The main goal of applying regenerative medicine to clinical practice is to restore function by providing viable cells based on the use of a novel therapeutic approach to generate biological substitutes and improve tissue functions. Interest in corneal epithelium rehabilitation medicine is rapidly growing, given the exposure of the corneal outer layers to external insults. Here, we performed a review of basic, clinical and surgical research reports on regenerative medicine for corneal epithelial disorders, classifying therapeutic approaches according to their macro- or microscopic target, i.e., into cellular or subcellular therapies, respectively.

## 1. Introduction

Regenerative medicine is an avant-garde perspective of the 20th century. This breakthrough and ever-evolving discipline has drastically changed diagnostic and therapeutic approaches to pathologies involving various systems and with different etiopathogenetic mechanisms. The major fascinating prospect of this research field is the ability to reconstitute and repair damaged tissues with non-specialized grafts, which are capable of differentiating into varieties of organ-specific cells. Therefore, these new cells could replace in loco the impaired ones. Regenerative medicine is considered the cornerstone upon which therapeutic strategies of the future will be based, and it is an evolving field in ophthalmology as well; for instance, in 2017, the FDA approved the first gene therapy treatment for Retinitis Pigmentosa (LUXTURNA^®^). However, the applications of regenerative medicine in ophthalmology are not confined to the posterior segment [1,2]; indeed, economic and scientific resources have been invested for years into providing effective strategies that can also restore and manage disorders involving the anterior segment of the eye. Being the human window to the world, the cornea is the outermost layer of the human eye, constantly exposed to thermal, chemical and physical insults. Flaxman et al. have shown that about 10% of people with visual acuity impairment are affected by alterations in corneal clarity and that corneal issues are the fifth cause of visual acuity impairment and blindness worldwide [3]. Still, the current gold standard treatment for corneal diseases that severely compromise visual acuity is corneal transplantation. Nevertheless, the number of corneal transplantations is meager because of political, economic and socio-sanitary issues, in addition to the scarcity of donors and suitable corneal tissues [4,5]; in clinical reality, about 80% of grafts are discarded due to inadequate harvesting and/or storage [6,7]. This scenario highlights how it is and how it will be increasingly necessary to stimulate current corneal investigation to benefit from cell therapies and not tissue grafts. Basic corneal research, in an increasingly fine-tuned and accurate combination with genetics and biochemistry, could allow reactivation and/or in situ replacement of damaged corneal cells with non-invasive or minimally invasive techniques, enabling multimodal management of corneal disorders by taking advantage of innovative laboratory techniques and in a faster timeframe than expected for surgical transplantation. Therefore, the emphasis on new regenerative therapeutic options is considerable and increasingly growing in an attempt to find a viable alternative for a larger proportion of patients. In light of this, we decided to focus this review on the current emerging strategies and future perspectives of regenerative management of corneal diseases, especially concerning the corneal epithelium.

## 2. Epithelial Corneal Regeneration: An Overview

The corneal epithelium represents 10% of the total tissue thickness, and it is composed of at least six layers: two superficial or wing cells, three to five layers of intermediate cells and the last layer of basal cells. Comparing the epithelium with the stroma and endothelium, it heals through the proliferation of stem cells, their differentiation into specialized epithelial cells and their migration to the damaged area. Otherwise, central epithelial cells can repair slight injuries on their own. In contrast, stromal cells undergo transformation, whereas the endothelium regenerates especially via cell migration [8]. Several mechanisms of action and molecules that finely regulate these cellular functions come into play in the regeneration process. This intricate network explains the complexity of this phenomenon and how to manage it. Injury-induced intercellular crosstalk has been proposed, in which the most relevant growth factors involved are insulin-like growth factor (IGF-1), epidermal growth factor (EGF), hepatocyte growth factor (HGF), keratinocyte growth factor (KGF), transforming growth factor-β (TGFβ) and cytokines, such as interleukin 1 (IL-1) and tumor necrosis factor α (TNF-α) [9], in particular triggering cell migration [10,11]. Additionally, this network interfaces with a non-static but dynamic microenvironment, comprising the extracellular matrix (ECM). It acts as a scaffold in everlasting remodeling modulation of cellular proliferation, migration and differentiation [12,13,14,15,16,17,18,19,20,21]. Corneal epithelial cells secrete components forming ECM above Bowman’s layer. All of these are essential players in ensuring tissue homeostasis and repair; if any one of these is deficient, the critical balance breaks down. The daunting challenge in regenerative medicine is to devise specific therapies to manage each component of this cascade that is altered, from the macroscopic tissue to the microscopically altered gene. In this review, the state-of-art therapies are reported in a bio-pathological overview, that is, beginning from how to manage macroscopic cellular alterations (stem cells and related surgical techniques) to microscopic intercellular and subcellular signaling alterations (use of exosomes, genes therapy and regulatory RNAs). The main therapeutic targets of regenerative ophthalmology are shown in Figure 1.

## 3. Cell Therapies

The cell is the basic biochemical–physical unit of any tissue, and the damage exhibited on a specific tissue is secondary to injury exerted on its cells: cells suffer and die, and consequently so does the tissue. Therefore, pioneering studies in terms of regenerative medicine focused on replacing damaged cells through the transplantation of stem cells. Regarding the regeneration of corneal epithelium, mainly limbal corneal stem cells, mesenchymal cells and others of non-corneal derivation have been and still are employed and are discussed below along with the main surgical techniques involved.

### 3.1. Limbal Stem Cells

The implementation of stem cells has been, and still is, a milestone in treatments in terms of regeneration. Concerning the cornea, epithelial cells originate from stem cells, continuously reconstituting tissues. The epithelial corneal stem cell reservoir exists in the limbal niche. In this location, stem cells are quiescent and remain in this condition for long periods, although they have oligopotency properties and self-renewal such that corneal homeostasis is maintained. Limbal niche comprises two different structures proposed to be stem cell reservoirs: the palisades of Vogt [22] and the limbal epithelial crypts [23]. In addition to corneal stem cells, the limbal niche harbors other cell types, including limbal stromal fibroblasts, melanocytes, Langerhans cells and early transient amplifying cells (TACs) [24]. It has been shown that limbal stem cells give rise to TACs, which in turn proliferate and give rise to differentiated and functional tissue cells. These final cells are the end product of the regeneration process, have little proliferative capacity and do not self-renew; they migrate and replace the damaged ones [25]. Whatever the triggering agent (congenital or acquired), should this reservoir be damaged, the epithelium can no longer self-renew, resulting in limbal stem cell deficiency (LSCD) (Figure 2) [26,27,28,29]. One of the greatest challenges scientific research has encountered in investigating this area is identifying markers specific to stem cells in order to be able to recognize and use them accordingly. Many markers have been investigated, but currently there is a lack of specific factors capable of labeling these cells. The most promising was certainly p63+, a transcription factor, proposed by Pellegrini et al. in 2001 [30], also supported by the study of Rama et al. [31]. Ksander et al. demonstrated that ABCBA5 can be used as a limbal stem-cell marker, generating ABCB5 knockout mice, which lost regeneration and repair abilities [32]. This evidence has been confirmed by other studies conducted on humans [33,34,35]. These characteristics have hindered the development of techniques for isolating, growing and transplanting them to manage corneal epithelial diseases. However, despite the limitations related to corneal transplantation and the use of differentiated cells to repair corneal damage, several studies have investigated innovative strategies to improve the use of limbal stem cells by virtue of their proliferating and repairing properties and selectively transplant them.

Limbal stem-cell transplantation is often necessary for vision-threatening LSCD and in eyes refractory to medical treatments aiming to optimize the ocular surface and promote epithelial proliferation. Over the past 30 years, various surgical procedures have been reported [36], and a treatment algorithm based on the stage of the disease was proposed by the International Limbal Stem Cell Deficiency Group [37]. Eyes in which vision is significantly impaired often require surgical treatment to restore the ocular surface. The following paragraphs describe the main surgical techniques performed to date.

#### 3.1.1. Conjunctival–Limbal Autograft (CLAU)

The conjunctival–limbal autograft consists of conjunctival and limbal transplantation of tissue from the healthy fellow eye, including Vogt’s palisades [38]. The use of CLAU for treating unilateral LSCD was first described by Kenyon and Tseng [39] with a case series of eyes presenting acute and chronic LSCD. This technique was the treatment of choice in patients with unilateral LSCD for over two decades, and 35% to 88% of eyes treated showed an improvement in BCVA of two lines or more [40]. To avoid the risk of conjunctival tissue invasion of the cornea over the areas of exposed corneal limbus where there is no barrier function, Chan et al. combined the advantages of the CLAU technique with the barrier effect derived from a cadaveric keratolimbal autograft (KLAL) [41] in patients with severe chemical/thermal injuries, resulting in sustained ocular surface stability and avoiding the invasion from the host conjunctiva of the exposed limbal and corneal areas. Cheung et al. later compared the outcome of the KLAL procedure with the living related conjunctival limbal allograft (lr-CLAL), where allogenic limbal and conjunctival tissues are harvested from a matched living relative [42]. This latter procedure demonstrated lower rejection rates and improved graft survival compared to KLAL [43]. The favorable outcome of CLAU, mainly due to safety and long-term efficacy, and the histocompatibility between graft and host [44] contrasts with the potential donor eye limbal stem-cell failure secondary to the need for resecting 40–50% of host limbal stem-cell population [38,45]. A “minimal CLAU” procedure technique has been proposed, but the long-term efficacy and safety are unknown [46].

#### 3.1.2. Ex Vivo Cultivated Limbal Epithelial Transplantation (CLET)

To overcome the need for a large amount of donor tissue, a novel ex vivo cultivated limbal epithelial transplantation (CLET) was proposed by Pellegrini et al. [47]. A small portion of limbal stem cells is collected from the healthy fellow eye (auto-CLET) or, in case of bilateral LSCD, from a living related donor or cadaveric unrelated donor (allo-CLET) [48,49]. Limbal cells are cultured and expanded in vitro using an amniotic membrane [50] or a tissue-engineered scaffold [51,52]. The anterior lens capsule (ALC) can be easily obtained during cataract surgery and, due to the optical transparency and widespread availability, proved to be an excellent scaffold for limbal stem cell proliferation and ocular surface restoration [53,54]. CLET shows fewer risks for the donor eye, avoiding the risk of running into iatrogenic LSCD, and adding the possibility to treat partial bilateral LSCD [55]. Amniotic membrane (AM) carriers also have intrinsic anti-microbial, anti-fibrotic and anti-inflammatory properties [56], increasing the healing function of CLET. In 2010 Rama et al. demonstrated the long-term favorable outcome of this procedure, where 76.6% of the grafts remained stable after 10 years of follow-up [31]. The main disadvantage is the required laboratory facility and trained staff for stem cell culturing.

#### 3.1.3. Simple Limbal Epithelial Transplantation (SLET)

First described by Sangwan et al. in 2012, this novel technique was used in six patients with unilateral LSCD and showed several advantages in terms of safety and effectiveness [57]. A small limbal tissue graft from the contralateral eye (auto-SLET) or from a donor eye (allo-SLET) is divided into smaller pieces and allocated over the recipient cornea, using amniotic membrane to support the growth of epithelial cells. SLET combines the advantages of being an easily affordable, one-step procedure that requires minimal laboratory assistance and donor tissue [57]. Results from a multicenter study indicate that autologous SLET is an effective and safe modality for the treatment of unilateral LSCD, where 57 of 68 eyes that received auto-SLET (83.8%) reached the main outcome of a complete restoration of the corneal epithelium [58], with a similar success rate and visual acuity improvement to those reported with CLET. Allo-SLET represents a valid option for bilateral LSCD such as persistent epithelial defects [59] and iatrogenic LSCD [60]; however, it requires indefinite use of systemic and topical immunosuppressants [61]. A recent retrospective case series compared the outcome of patients treated with amniotic membrane alone vs. AMT + allo-SLET, highlighting the superior outcomes of the latter method in terms of time of recovery and rate of side effects [62].

#### 3.1.4. Holoclar^®^ (Ex Vivo Expanded Autologous Human Corneal Epithelial Cells Containing Stem Cells)

Holoclar was the first approved stem cell-based therapy in medicine, used to treat eyes with moderate to severe limbal stem-cell deficiency [63]. Healthy limbal tissue (1–2 mm) is collected from the unaffected eye, grown in a laboratory on a fibrin matrix and transplanted into the injured eye to restore the stem-cell population. Efficacy depends on the presence of LSCs in the product, and the content of these cells can be determined from immunostaining for the p63 marker. Holoclar was developed as an orphan medicine based on the results of two retrospective studies [31,47] and is now under conditional authorization in Europe; marketing authorization in the EU requires submission of a comprehensive product dossier to the EMA and to the Committee for Advanced Therapies (CAT). To ensure Holoclar is used safely, a Risk Management Plan (RMP) has been developed [64]. Holoclar is now under approval in the US (phase 4 clinical trial; ClinicalTrial.gov NCT02577861).

### 3.2. Mesenchymal Stem Cells

Epithelial stem cells are not the only ones available as a regenerative reservoir at the limbal site; indeed, another stem-cell population exists, and it is represented by mesenchymal stem cells (MSCs). These are adult stem cells characterized by PAX6 and ABCG2 expression [65] that give rise to different mesenchymal and non-mesenchymal cell lineages, such as fat, bone, skeletal muscle, cerebral tissue and corneal layers [66]. In the latter, corneal stromal MSCs can differentiate into keratocytes [67] as well as hinder the formation of extensive and disabling leucomas in terms of visual acuity. Corneal transparency is decisive for guaranteeing and maintaining long-term visual restoration after injury [24,68]. They are peculiar and attractive alternatives due to their high proliferative potential, self-renewal and ability to be differentiated into several cellular types. Moreover, these cells modulate stem-cell migration to selected damaged sites [69,70] and immune regulation [70,71]. Nevertheless, the MSCs themselves can migrate and recreate a regenerative microenvironment through the release of soluble factors [72]. To date, the most studied and effective cells are bone marrow-derived MSCs (BM-MSCs) and human adipose-tissue-derived MSCs (AT-MSCs). BM-MSCs were at first identified as the most promising source of MSCs. The allogenic transplantation of this lineage on the human ocular surface was first described in a randomized clinical trial in 2019 [73], but their cell harvesting and isolation is more challenging rather than that of AT-MSCs [74]. The latter are currently considered the most suitable for repairing corneal epithelial damage [75,76,77] because they are easily applied in vivo from peripheral adipose tissue suction [78]. Therefore, this lineage of MSCs seems to represent a safer alternative to BM-MSCs in patients with severe bilateral LSCD [78]. Another advantage of AT-MSCs is their location in considerable quantities in human tissues: this allows their autologous transplantation [79].

Despite this encouraging evidence, MSCs still have significant downsides. Since they can give rise to different cell types, the variables of differentiation into corneal epithelial cells are manifold, and the differentiation process is still in part unknown. Indeed, for instance, mesenchymal stem cells cultivated on human amniotic membrane (AM) resulted in restoring the corneal epithelial phenotype damaged by LSCD, although it remains unclear whether MSCs can transdifferentiate into corneal epithelial cells [80]. Moreover, it is necessary to recreate the microenvironment in which the native limbal stem cells reside: each step is crucial to ensuring the success of the process. In addition, a long in vivo follow-up is required [81]. Another disadvantage is the choice of an eligible donor organism: advanced age, unfavorable metabolic conditions and genetic inheritance may negatively affect the tissue proliferation, differentiation and thus regeneration [82,83]. Given the high replication rates, an increased risk of tumor development cannot be excluded [84]. Few studies have been conducted to date on in vivo implantation of such cells. It is a prerogative of scientific research to standardize the processes of harvesting and use in order to ensure few risks and maximum results.

### 3.3. Non-Corneal Stem Cells

As described previously, the existence of stem cells in other areas of the human body besides the limbus is known. In case of severe bilateral LSCD, the only option is to perform an allogenic transplantation, with the burden of a prolonged postoperative immunosuppressive therapy. To avoid this scenario, the transplant of autologous tissue of non-ocular origin has been studied as an alternative to replace injured corneal tissue [56]. The leading surgical techniques are reported below.

#### 3.3.1. Autologous Ex Vivo Cultivated Oral Mucosal Epithelial Cells (COMET)

The first use of a cultured non-limbal autologous cell type to treat bilateral LSCD was reported in 2004 by Nakamura and al. [85] for six patients with severe bilateral ocular surface disease. COMET collected from a healthy oral mucosa and placed in patients with Steven Johnson syndrome (SJS) and severe burn injury demonstrated a minimal risk of graft rejection; 70.8% of eyes treated achieved a corneal epithelium restoration, and 63.5% had an improvement of at least two lines in the visual acuity [86]. A long-term follow-up of 19 patients with severe ocular surface disorders treated with COMET showed sustained reconstruction of the ocular surface epithelium in many eyes with severe OSD [87], with a small persistent epithelial defect noted in only 5–26% of the eyes during follow-up period.

#### 3.3.2. Hair Follicle Stem Cells (HFSCs)

Blazejewska et al. in 2009 reported that stem cells from hair follicles could be effectively differentiated into corneal epithelial-like cells when cultivated in a media conditioned by limbal fibroblasts [88] and effectively re-epithelialize the corneal surface [89].

#### 3.3.3. Epidermis

Stem cells isolated from the skin of goats and transplanted onto goats with severe LSCD could differentiate into corneal epithelial cells and regenerate a transparent corneal epithelium [90].

The surgical procedures concerning cell transplantation are summarized in Figure 3.

## 4. Subcellular Therapies

Whereas cells are the biochemical–physical units of tissues, they themselves represent an intricate and complex microcosm. Scientific research has shifted emphasis toward the inter- and intracellular mechanisms involved in cell regeneration in order to enable even more sophisticated and targeted therapeutic management, thus developing subcellular therapies. To date, the most promising in the corneal area are the application of extracellular vesicles (EVs), gene therapy and regulatory RNAs, particularly miRNAs.

### 4.1. Exosomes

Exosomes belong to the family of extracellular vesicles (EVs). Their membrane is made up of double-layer lipids, and they are round organelles about 40–100 nm in size [91]. In 1980, they were defined as reticulocyte formation products [92]. It was originally supposed that these organelles excreted waste products from the cells into the extracellular space [93]. Otherwise, over the years they have aroused particular interest: it has been hypothesized that the role of exosomes is to carry out the cell-to-cell communication of most cells, including MSCs. Recently, this peculiar activity of theirs has gained the attention of researchers. Indeed, MSCs secrete exosomes by triggering paracrine signaling on other cells [72,94]. Concisely, cells release these vesicles composed of different intracellular compounds such as nucleic acids, proteins and lipids, each one providing a specific biological function depending on the recipient cell [95]. The membrane of these EVs is also composed of proteins, which are considered to play a key role in intercellular signal transduction. Therefore, it is supposed that exosomes are intercellular mediators and transmitters of information that are crucial for the performance of cellular physiological and pathological mechanisms, such as inflammation, angiogenesis and immunomodulation. However, how does their cargo reach its final destination? The content inside the exosomes must be incorporated by the recipient cell. One of the best-known processes is endocytosis [96], but exosomes may also fuse with the membrane of the recipient cell and release their contents directly into it [97], or the two membranes may interact and establish a receptor-ligand duo [98]. Once embedded intracellularly, the cargo is enabled to perform its action. Regardless of the signal transduction pathway, it has been suggested that this intercellular communication network could be emphasized from a therapeutic perspective, in particular for the delivery of drugs, genes or other agents, that may, for instance, induce a proliferative stimulus or gene expression. Angiogenetic and regenerative action in cutaneous wound repair [99] and anti-apoptotic activity in retinal cells have been demonstrated [100]. Although currently the role of exosomes in ocular pathophysiology is still elusive, their expression could also potentially be applied in the ocular and corneal context. In support of this, a bidirectional flow of exosomes between corneal epithelial and stromal cells was observed in vitro [101,102]. In this regard, many studies have documented the effects of EVs in corneal regeneration, both in vitro [101,102] and in vivo [103]. Indeed, it has been shown that they are able to accelerate wound healing [101] and modulate fibrosis [102,103] in damaged mouse corneas. The main studies on this topic are shown in Table 1.

Even though further in vivo studies are needed, accumulating evidence attests to the role of exosomes as cell-to-cell intermediaries in the regulation of various mechanisms that may occur within the corneal tissue. According to reports, one of the most promising functions from a therapeutic perspective would be to use them as conveyors of coding material, i.e., nucleic acids, that can be incorporated and expressed by the target cell and change and replace its abnormal or missing properties. The mechanism of MSC-induced EVs is illustrated in Figure 4.

### 4.2. Gene Therapy

Gene therapy involves the transfer of a given gene into target cells in order to modify their genetic heritage and restore the cellular alterations found. Given its immune privilege, avascularity and easy accessibility, cornea can be considered a suitable tissue for the application of this therapeutic strategy. In essence, the challenge of this procedure is to design the appropriate carrier for this purpose. Two types of carriers have been identified: viral carriers and nanoparticles. Concerning the former, adenoviruses and lentiviruses have been the most widely used in corneal tissue. The latter proved to be more effective in terms of expression of the carrier gene than nanoparticles. On the other hand, the latter are easier to manipulate in laboratories and have a reduced rate of inflammation and infection [10]. This therapeutic strategy is already undergoing experimentation for the management of some inherited metabolic diseases with corneal involvement, such as mucopolysaccharidosis. They comprise a group of disorders marked by lysosomal accumulation of undegraded glycosaminoglycans (GAGs) due to genetic mutations encoding lysosomal hydrolytic enzymes. The most severe of all is mucopolysaccharidosis type 1 (MPS1), also known as Hurler syndrome, caused by a defective form of alpha-L-iduronidase (IDUA). Affected children show cognitive deficits and ocular changes affecting visual acuity, including corneal clouding: this is thought to be due to the presence of stromal cells engorged with vacuoles containing abnormal GAGs and altering the usual organization of collagen fibrils and stromal geometry [104]. The current systemic therapeutic options have not shown significant efficacy at the corneal region, and transplantation has been shelved due to the high rejection rates recorded in past years in these children. Although it is still a pre-clinical project, Vance et al. studied the transduction of adeno-associated virus (AAV) IDUA gene addition therapy in MPS1 fibroblasts: they observed the restoration of IDUA function, and this could contribute to the corneal clarification [105] (to be verified in future studies). Considering these data, another area of application could be genetic corneal epithelial diseases, such as the autosomal dominant epithelial recurrent erosion dystrophies (EREDs). This family of corneal dystrophies is characterized by painful recurrent epithelial erosions leading to central corneal opacification with visual impairment in half of the patients [106]. They typically begin in childhood and persist throughout life, alternating between active and silent phases. Causing the disease is a heterozygous missense mutation in the COL17A1 gene, coding for collagen XVII [107,108]. Conveying the healthy gene in corneal epithelial cells, stem and already differentiated, could be a potential alternative to treating these dystrophies. Underlying gene therapy is, evidently, the imperative requirement to perform preventive genetic mapping at least concerning the genes implicated in corneal homeostasis, giving special attention to allelic variants and the corresponding variables in terms of expression and penetrance in the short- and long-term of the corneal pathologies encountered.

### 4.3. miRNA

MicroRNAs are noncoding short-chain RNA sequences. They do not determine protein production but participate in protein translation indirectly. They are to be regarded as regulators of protein expression, as they bind specific mRNA molecules promoting or hindering the synthesis of the proteins they encode. Their role in immune regulation, differentiation, angiogenesis and neovascularization has been described [10]. They represent key epigenetic modulators of cellular activities, including ocular ones [109]. Regarding corneal epithelium regeneration, both inhibitor and tissue repair-promoting miRNAs have been documented. For instance, miR-205 promotes the spread of epithelial cells to the site of corneal damage upregulating the AKT- and F-actin-mediated pathways [110] and inhibiting the KCNJ10 channel [111]. Moreover, miRNA-143-3p inhibition downregulates the expression of α-SMA, thereby reducing contractility of myofibroblasts and thus the tissue fibrotic response [112]. MiRNA-200a also inhibits fibrosis blocking corneal epithelial cell migration [113]. The therapeutic application of these regulators is made complex by their concomitant modulation of activities of several targets with different functions and may therefore go on to alter pathways other than the desired ones. On the other hand, this multi-targeted pattern setting could be an attractive option to deal with a process as intricate as tissue repair: it would be beneficial to develop selected miRNAs and antagomirs to modulate corneal homeostasis [10], as already studied in herpetic stromal keratitis [114] and other medical disciplines [115,116]. There are also systemic conditions that affect corneal regeneration, such as diabetes: it is known that this metabolic disorder delays and impairs corneal repair [117]. In this regard, the insertion of specific miRNAs has been shown to promote corneal regeneration by increasing the expression of c-MET and inhibiting that of cathepsin F and MMP-10, thus accelerating tissue healing in diabetic patients [118]. In addition, miRNA 146-α can modulate corneal regeneration and stem reservoir maintenance [119,120] and it has been shown that its overexpression can alter normal repair functions in diabetic corneas [121,122]. The main studies regarding the role of miRNAs in corneal regeneration are summarized in Table 2.

In light of this, a potential perspective would be to exploit the intercellular interchange of certain miRNAs mediated by MSC-derived EVs to specifically orchestrate target cells. In support of this, miRNAs have been identified that are expressed in high amounts in stem corneal cells as opposed to differentiated corneal cells, which could be integral elements of the processes of stemness maintenance but also of subsequent differentiation [123].

## 5. Conclusions

Vast and exciting advancements have been made over the years in the field of regenerative medicine in ophthalmology, and many innovative and promising perspectives are emerging. On the other hand, clinical applications are still limited and require appropriate regulation: a single treatment is not enough, and it is necessary to develop a long-term care-project that constitutes a well-established all-round therapeutical approach. Currently, the aforementioned strategies are the most important and promising in the corneal area. The early stages of this scientific research were focused on the direct use of stem cells, which has revolutionized the therapeutic approach of standard medicine. In recent years, researchers have shown an increasing and enthusiastic interest for regenerative medicine nano-techniques, particularly MSC-induced exosomes and EVs (Figure 5). They may form a futuristic and minimally invasive therapeutic strategy, being able to act as biological carriers with minimal risks of infection and maximum advantages in terms of in vivo application and handling over tissue grafts and the cells themselves.

In our opinion, the most promising fields for future application of these nanotechnologies could be metabolic disorders, genetic diseases and traumatic (physical, chemical, etc.) corneal damage. However, the few available treatments are too expensive to be accessible to all. It may be unnecessary to highlight this; nevertheless, their application requires the stipulation of rigorous guidelines to properly select suitable patients and to protect them from therapeutic abuse for experimental and economic purposes, which is lacking to date. Not only the costs but also the complexity of these avant-garde technologies make their use in clinical practice far more arduous than expected. This highlights the pressing urgency to educate and train specialized professionals who are liaisons between the corneal clinic and the far more intricate and articulated pathophysiological substrate. Therefore, the establishment of a dedicated ophthalmology researcher, an “ocular biologist”, could reasonably meet this need as a professional liaison between basic and cutting-edge research as well as with the clinical ophthalmologist. This specialist could manage both the issues and potential of a given innovative technique for the treatment of a specific pathology. Moreover, as is evident from our report, the need for the integration of various technologies (e.g., MSC-induced exosomes and gene therapy, EVs and miRNAs) appears increasingly evident. A multidisciplinary (clinical, biological and genetic) approach is also of crucial importance in conceiving further promising and innovative individualized therapeutic strategies.

## Figures and Tables

**Figure 1 ijms-23-13114-f001:**
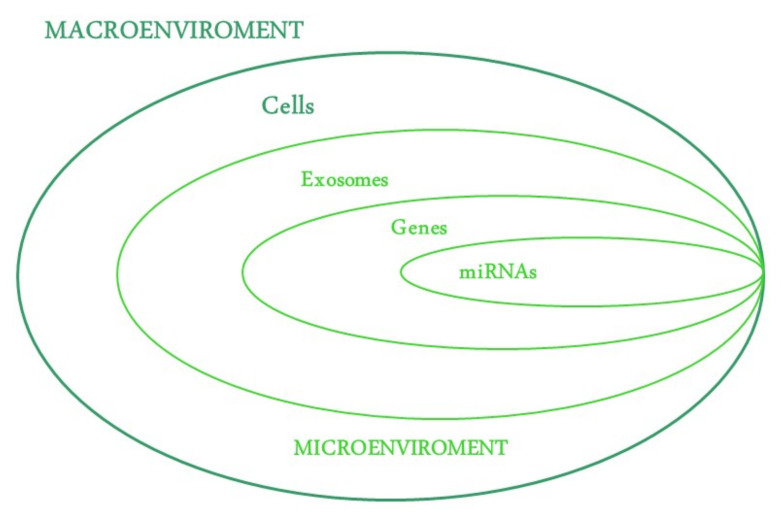
Targets of Regenerative Ophthalmology: The diagram illustrates the main targets of regenerative therapies: cells, as a complex macroscopic unit, and the microscopic constituents embedded within them.

**Figure 2 ijms-23-13114-f002:**
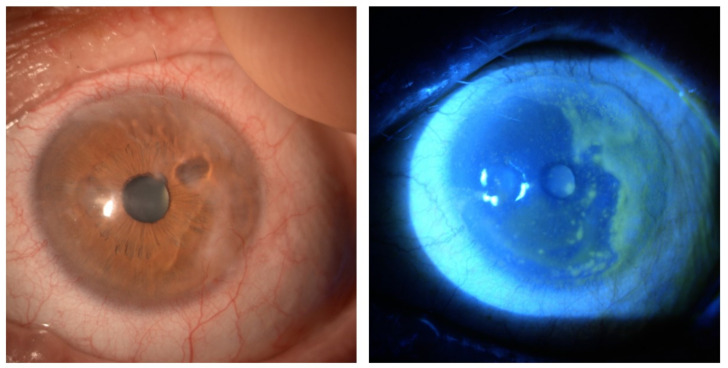
A case of limbal stem-cell deficiency (LSCD) in a patient with chronic graft-versus-host disease (GVHD).

**Figure 3 ijms-23-13114-f003:**
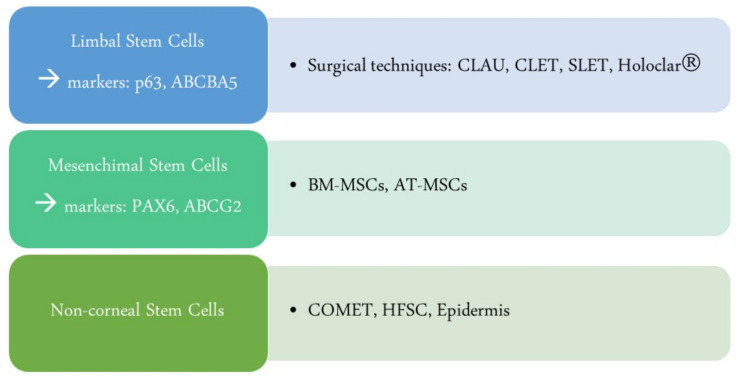
Corneal stem cell transplantation: This scheme summarizes the current main surgical techniques to replace limbal stem cells. CLAU: Conjunctival–limbal autograft; CLET: ex vivo cultivated limbal epithelial transplantation; SLET: simple limbal epithelial transplantation; Holoclar^®^, ex vivo expanded autologous human corneal epithelial cells containing stem cells; BM-MSCs: bone marrow-derived mesenchymal stem cells; AT-MSCs: adipose-tissue-derived mesenchymal stem cells; COMET: autologous ex vivo cultivated oral mucosal epithelial cells; HFSCs: hair follicle stem cells.

**Figure 4 ijms-23-13114-f004:**
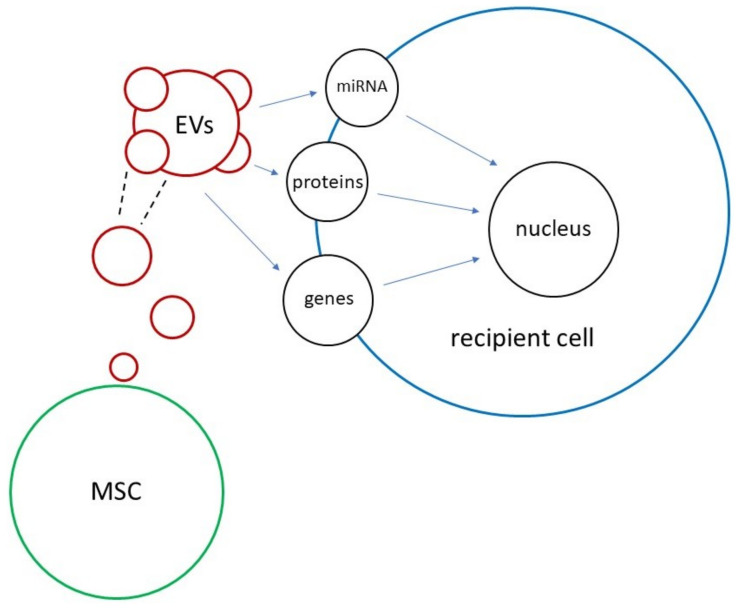
EV-mediated intercellular communication: This diagram shows the mechanism of intercellular communication mediated by MSC-induced exosomes. MSCs: mesenchymal stem cells; EVs: extracellular vesicles.

**Figure 5 ijms-23-13114-f005:**
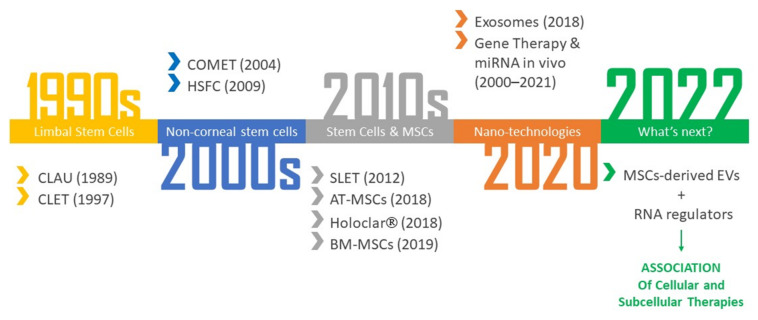
Historical development of corneal regeneration. This diagram summarizes the different steps and goals of corneal regeneration over the years. CLAU: Conjunctival–limbal autograft; CLET: ex vivo cultivated limbal epithelial transplantation; COMET: autologous ex vivo cultivated oral mucosal epithelial cells; HFSCs: hair follicle stem cells; SLET: simple limbal epithelial transplantation; AT-MSCs: adipose-tissue-derived mesenchymal stem cells; Holoclar^®^, ex vivo expanded autologous human corneal epithelial cells containing stem cells; BM-MSCs: bone marrow-derived mesenchymal stem cells; EVs, extracellular vesicles; siRNA, silent interfering RNA, miRNA, microRNA.

**Table 1 ijms-23-13114-t001:** EVs roles in corneal regeneration in mice.

MSC-induced EVs stimulate corneal wound healing in vitro.	Samaeekia et al. [101]
EVs-induced α-SMA increases contractile capacity of myofibroblasts.	McKay et al. [102]
Specific siRNA expression can cause defective packaging of EVs. Therefore, this prevents miRNA incorporation into exosomes, causing a prominent hindrance to corneal regeneration.	Shojaati et al. [103]

MSCs: mesenchymal stem cells; EVs, extracellular vesicles; α-SMA, alpha-smooth muscle actin; siRNA, short interfering RNA; miRNA, microRNA.

**Table 2 ijms-23-13114-t002:** miRNA roles in corneal regeneration.

MiRNA-205 promotes the spread of epithelial cells to the site of corneal damage (stimulating AKT- and F-actin-mediated pathways).	Yu et al. [110]
MiRNA-205 promotes the spread of epithelial cells to the site of corneal damage (inhibiting KCNJ10 channel pathway).	Lin et al. [111]
MiRNA-143-3p inhibition downregulates α-SMA, inhibiting a fibrotic response in damaged corneas.	Zhang et al. [112]
MiRNA-200a blocks corneal epithelial cell migration.	Luo et al. [113]
MiRNAs modulate wound healing, increasing c-MET expression in diabetic corneas.	Kramerov et al. [118]
MiRNAs modulate wound healing, inhibiting cathepsin F and MMP-10 expression in diabetic corneas.	Kramerov et al. [118]
MiRNA146-α levels regulate corneal regeneration and stem reservoir maintenance in diabetic patients.	Funari et al. [119] Poe et al. [120]
Downregulation of miRNA146-α repristinates adequate repair functions in diabetic corneas.	Winkler et al. [121] Poe et al. [122]

AKT, Ak strain transforming; KCNJ10, potassium inwardly rectifying channel subfamily J member 10; c-MET, mesenchymal–epithelial transition factor; MMP-10, matrix metalloproteinase 10; α-SMA, alpha-smooth muscle actin.

## Data Availability

Not applicable.

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
