# Peer review of "Corneal Epithelial Regeneration: Old and New Perspectives"

_ijms, 2022, doi:10.3390/ijms232113114_

Round 1

Reviewer 1 Report

The authors presented a review on corneal epithelial regeneration, which is interesting and can provide guidance to our clinical work. I just have a few comments on the manuscript:

1.       Anterior lens capsule can also be used to culture epithelial cells (PMID: 32914585), the author can make a brief discussion under section 3.1.1.2.

2.       I suggest the authors to give the explanation of the abbreviations in the figure legends.

3.       There are several typo or grammatical errors throughout the manuscript, please proofread the manuscript carefully.

Author Response

Dear reviewer, Thank you for your time and your suggestions. 
Please find our replies to your comments below.

  1. Anterior lens capsule can also be used to culture epithelial cells (PMID: 32914585), the author can make a brief discussion under section 3.1.1.2

 Response: we thank the reviewer for the timely suggestion. An additional comment has been added to the section 3.1.2., where we included ALC as a substrate for epithelial cells culture.

  1. I suggest the authors to give the explanation of the abbreviations in figure legends.

Response: Explanations to the abbreviations have been added to the figure legends.

  1. There are several typos and grammatical errors throughout the manuscript, please proofread the manuscript carefully.

Response: The manuscript has been proofread thoroughly.

Kind Regards,

The Authors.

Reviewer 2 Report

Please read the attached file.

Thank you.

Author Response

Dear reviewer, thank you for your help and your suggestions. We tried to modify our work according to your advices. Please find below our replies.

- In the introduction section, you should emphasize the research gaps and explore potential areas in a particular field (Section 1). Please revise the meaning of this sentence “In light of this, we decided to focus this review on the present-day and future perspectives of regenerative management of corneal diseases, especially concerning the corneal epithelium.”

Response: we thank the reviewer for the timely suggestion. We modified the introduction section following the reviewer’s advice.

- Figure 1: Please rewrite its title. It should not be a sentence, and it would be a noun phrase(s).

Response: we rewrote the title of this figure.

- There is only one subsection 3.1.1. Surgical Techniques without other subsections 3.1.2 and 3.1.3 … etc. Please consider removing it.

- There is only one subsection 3.2.1. BM-MSC and AT-MSC transplantation without other subsections 3.2.2 and 3.2.3 … etc. Please consider removing it.

- There is only one subsection 3.3.1. Non-Limbal Epithelial Stem Cell Transplantation (non-LESC transplantation) without other subsections 3.3.2 and 3.3.3 … etc. Please consider removing it.

Response: we removed the sections 3.1.1., 3.2.1. and 3.3.1. following the reviewer’s suggestions.

- Figure 4: the quality of the figure is poor. Please increase the resolution of this figure.

Response: we increased the resolution of this figure.

- Lines 415-419: please rewrite this sentence (“To date, the aforementioned strategies are the most important and the most promising in the corneal area: certainly, since the early stages of scientific research focused on the direct use of stem cells, which revolutionized the therapeutic approach of standard medicine, scientific research is reserving increasing and enthusiastic interest for regenerative medicine nano-techniques, particularly MSCs-induced exosomes, and EVs”). The grammar and structure of this sentence seem to be wrong. Please revise.

Response: we rewrote this sentence revising its grammar and structure.

- Lines 431-436: please rewrite this sentence (“Moreover, as is evident from our report, there appears to be an increasingly clear evidence of the need for the integration of various technologies, e.g., MSCs-induced exosomes and gene therapy, EVs and miRNAs, etc., which in turn confirms the requirement for a multidisciplinary approach (clinical, biological and genetic) to find increasingly promising and innovative therapeutic strategies that are more and more individualized and designed ad personam”). It seems to be wrong in grammar and structure. Besides, the sentence is too long and complicated for readers to understand.

Response: we rewrote the sentence above. We think its meaning is easier to understand now.

- Please make additional short review tables to list the contributions of the previous research instead of that listed one by one as presented in the text. It isn't easy to understand what the authors want to mention. Also, please give additional charts/ diagrams to summarize the historical development of the research field/areas or previous findings and then provide the trends for further study.

Response: we added Table 1. and Table 2. to summarize the roles of Extracellular Vesicles (EVs) and microRNAs respectively. In addition, we created a diagram to represent the stages of corneal regeneration over time and the possible future approaches in this field.

- Please provide clarity, novelty, and contribution to the research area for the review papers. It demands a robust, in-depth understanding of the subject and a well-structured arrangement of discussions and arguments.

Response: we conducted a thorough review of scientific literature to provide a general, but accurate overview regarding innovative corneal regeneration techniques, which will be increasingly important in the management of corneal epithelial diseases.

- Citation in the text and references should be followed the journal template. This research has a shortage of citations. The reviewer suggests that you should search the International Journal of Molecular Sciences or other Journals for more references that could be used to enrich your literature review. Generally, a manuscript (average) will have about 150-180 papers. References should be numbered in order of appearance and indicated by a numeral or numerals in square brackets—e.g., [1] or [2,3], or [4–6]. In the text, reference numbers should be in square brackets [ ] and placed before the punctuation; for example, [1], [1–3], or [1,3]. For embedded citations in the text with pagination, use both parentheses and brackets to indicate the reference number and page numbers; for example, [5] (p. 10) or [6] (pp. 101–105).

Response: We checked and edited our references as recommended by the reviewer and the journal instructions.

- Please check the English of the whole manuscript carefully. Could you please rearrange the layout of the manuscript to make it more coherent and logical?

Response: The manuscript has been proofread thoroughly.

- What are the pros and cons of this study? What is the main limitation of this study?

The pros of our review are the thorough review of the current literature on this topic, which provide a useful reference tool for understanding this intricate topic and the future perspectives in this field. On the other hand, to date, clinical applications are still limited; new future studies are needed to make these therapeutic strategies effective. We added these aspects in the conclusion paragraph.

- Could you please briefly conclude the field's trends for driving our future studies/approaches?

We reported in our conclusions the most suitable fields of application (genetic, metabolic, and traumatic corneal diseases) and the necessity of creating specialized professionals in handling these nano-technologies.

The reviewer hopes that his point of view could help the authors improve their work well.

I appreciate your work.

Response: Thank you again for your suggestions and your opinion.

Kind Regards,

The Authors

Reviewer 3 Report

Dear authors

Modern medicine strives not only for a temporary improvement in the patient but for a cure. The way to achieve this is through advanced therapy medicinal products. Unfortunately, in my opinion, modern science is only at the beginning of the road due to the high costs of treatment and the small number of preparations of proven quality. In relation to the topic of this paper, I would kindly ask you to refer in your paper to the Holoclar product and the EMA and FDA regulatory requirements for the therapies in question. Unfortunately, when discussing this, the authors should bear in mind the regulations; failure to comply with them often results in harm to patients. An example of this is the reported use of stem cells in peri-implantation. In my opinion, the paper should also devote at least one paragraph to the dangers of irresponsible use of new types of therapy due to profit motives or an unjustified belief in the success of the therapy.

Author Response

"Modern medicine strives not only for a temporary improvement in the patient but for a cure. The way to achieve this is through advanced therapy medicinal products. Unfortunately, in my opinion, modern science is only at the beginning of the road due to the high costs of treatment and the small number of preparations of proven quality. In relation to the topic of this paper, I would kindly ask you to refer in your paper to the Holoclar product and the EMA and FDA regulatory requirements for the therapies in question. Unfortunately, when discussing this, the authors should bear in mind the regulations; failure to comply with them often results in harm to patients. An example of this is the reported use of stem cells in peri-implantation. In my opinion, the paper should also devote at least one paragraph to the dangers of irresponsible use of new types of therapy due to profit motives or an unjustified belief in the success of the therapy."

Response: we thank the reviewer for taking the time to review our work and raising such an important issue. A discussion on Holoclar product was added with a brief description of the current EMA and FDA approval status (3.1.4). In addition, in the conclusion paragraph we mentioned the high costs of the procedures described and the possible abuse of them for economic and experimental reasons.

Round 2

Reviewer 2 Report

The authors have corrected and answered all my comments and questions. 

The reviewer suggests that the manuscript should be accepted for publication. 

Thank you.